# Realfood and Cancer: Analysis of the Reliability and Quality of YouTube Content

**DOI:** 10.3390/ijerph20065046

**Published:** 2023-03-13

**Authors:** Sergio Segado-Fernández, Ivan Herrera-Peco, Beatriz Jiménez-Gómez, Carlos Ruiz Núñez, Pedro Jesús Jiménez-Hidalgo, Elvira Benítez de Gracia, Liliana G. González-Rodríguez, Cristina Torres-Ramírez, María del Carmen Lozano-Estevan

**Affiliations:** 1Department of Health Sciences, Universidad Europea de Canarias, Calle Inocencio García, 1, La Orotava, 38300 Santa Cruz de Tenerife, Spain; sergio.segado@universidadeuropea.es; 2Faculty of Health Sciences, Alfonso X el Sabio University, Avda. Universidad, 1, Villanueva de la Cañada, 28691 Madrid, Spain; iherrpec@uax.es (I.H.-P.); eben@uax.es (E.B.d.G.); ctorrram@myuax.com (C.T.-R.); 3Department of Nursing, Human Nutrition and Dietetics, Universidad Europea de Madrid, Calle Tajo, s/n, Villaviciosa de Odón, 28670 Madrid, Spain; beatriz.jimenez@universidadeuropea.es; 4Program in Biomedicine, Translational Research and New Health Technologies, School of Medicine, University of Malaga, Blvr. Louis Pasteur, 29010 Málaga, Spain; carlos.ruiz@uma.es; 5Traumatology and Orthopedic Surgery Service, Hospital Universitario Nuestra Señora de Candelaria, Ctra. Gral. del Rosario, 145, 38010 Santa Cruz de Tenerife, Spain; pedrojh@outlook.com; 6VALORNUT-UCM (920030) Research Group, Department of Nutrition and Food Science, Faculty of Pharmacy, Complutense University of Madrid, 28040 Madrid, Spain; liligonz@ucm.es

**Keywords:** detection, health misinformation, healthcare professionals, public health, social media, cancer, diet

## Abstract

This study analyzes the quality and reliability of videos related to nutrition and cancer on YouTube. Study Design: An observational, retrospective, cross-sectional, time-limited study analyzing activity on the social network YouTube was proposed. Methods: The information from the videos was extracted through an API search tool, using the NodeXL software. The criteria to select the videos on YouTube were the keywords “real food”, “realfood”, and “cancer” and the hashtags #realfood and #cancer were present, videos in English and videos available on 1 December 2022. Results: The DISCERN value in the total number of videos viewed was 2.25 (±0.88) points, indicating low reliability. The videos uploaded by HRU represented only 20.8%. Videos suggesting that the use of foods defined as “real food” could cure cancer without the intervention of any other treatment accounted for 12.5%. Videos that provided external links to scientific/technical evidence verifying the information represented only 13.89% of the total number of videos. Of these videos, 70% corresponded to HRU. The DISCERN value for videos from HRU users was 3.05 (0.88), a value that reflects a good reliability of videos from these users. Conclusions: This study provides information on the content and quality of the videos that we can find on YouTube. We found videos of non-health users who do not base their content on any scientific evidence, with the danger that this entails for the population, but it also highlights that the videos published by HRU have greater reliability and quality, being better perceived by the population, so it is important to encourage healthcare professionals and health institutions to share verified information on YouTube.

## 1. Introduction

Cancer is currently one of the primary causes of death worldwide [1], with lung cancer being the fourth leading cause of death. When differentiating between countries according to their income, we find that it is in middle- and high-income countries where several types of cancer (lung, colorectal, and stomach) appear among the ten most probable causes of death [2]. About the evolution and development of cancer as a probable cause of death, if in 2020 a total of 19 million people were expected to be diagnosed with cancer [3], the Global Cancer Observatory of the World Health Organization estimates that it could cause a total of 28.9 million deaths worldwide in 2040 [4]. This data highlights the importance of this disease and the need for adequate information for patients, family members, and healthcare professionals [5,6].

The need to obtain information about a cancer diagnosis is so necessary for patients to know something such as treatments, side effects, or even how to live with cancer [7] through cancer survivor testimonies [8]. But it is necessary not to forget the mental well-being associated with the access to information; this access reduces anxiety, depression, and even frustration, and is something essential to improve the emotional and mental well-being of patients who have received a diagnosis of cancer, as well as their families [8,9]. Although, in many cases, the main source of information is healthcare professionals [10,11], it is true that doubts and questions sometimes cannot wait, at which point patients or family members seek information in the most accessible way possible, which is using the Internet [12]. 

It is important to note that on the Internet it is increasingly common to address social media for information [11,12], and social networks have emerged as tools that allow health-related content to be shared quickly and directly [13,14,15]. In particular, those of an audiovisual nature such as Instagram, TikTok, or YouTube are becoming increasingly important, as written health information is sometimes phrased in a way that is not easily understandable for people without adequate health literacy [16,17]. Among these, the social network with the largest number of users is YouTube [18], with an estimated 2.1 billion active users [19], generating 5 billion visits and 1 billion hours viewed every day [20,21].

Although social media represents an advantage when it comes to obtaining information about a given pathology due to the existing high quantity of information about health on multiples platforms such as blogs, webs [22], messaging apps (Telegram or WhatsApp) [23], there is a lack of control over the veracity and reliability of the health content shared by the users [22], thus representing a potential source of health misinformation [6,24,25,26] that can affect adherence to medical treatments and patients’ health [27].

Concerns about diet and food intake have increased in recent years to the current situation where 60% of the population admit to being worried about long-term risks of the food they eat [28]. This has led to a considerable increase in food-related searches on the various platforms available on the Internet [29,30,31]. One example is the concept of “real food”, a movement that promotes the consumption of fresh or minimally processed foods, or foods whose industrial or artisanal processing has not altered the quality of the natural properties of the ingredients [32]. One of the most prominent searches is related to the so-called superfoods. This type of food is erroneously associated with the cure of some diseases such as cancer [33]. In addition, information sources, which may include people who are not knowledgeable about nutrition, can be unreliable and contribute to the proliferation and dissemination of misinformation on nutrition-related topics [30,34].

In patients diagnosed with cancer, the role of nutrition and controlled diet is of great importance [35,36], and should be taken into account from the moment of diagnosis, representing an important part of the therapeutic process, and should therefore be applied in parallel with antineoplastic treatments [37,38]. Not forgetting those people who have survived cancer and need to maintain healthy dietary habits [39,40]. As patients’ need for nutritional information is so important [30], they even get to consume content on social media created by non-experts in nutrition [34]. These sources may even suggest the consumption of nutritional supplements as if they were foods and even proper treatments [28], always promoted as “anti-cancer,” “cancer-fighting,” or “cancer-busting” [30]. Although the intake of ultra-processed foods can be detrimental to health, and there is clear evidence of this [33,36,40], diet choices, nowadays, cannot be considered as an anti-cancer treatment. This type of misinformation can be particularly dangerous for patients diagnosed with cancer because it may be deemed as a real alternative [41,42], sometimes even leading to the abandonment of the prescribed medical treatment.

To the best of our knowledge, there are no previous studies that focus on the role of the real food movement and its association with cancer patients on YouTube. In this context, the aim of this study is to analyze the quality and validity of the existing videos on YouTube that relate the consumption of “real food” and cancer. Secondary objectives are: (i) to determine the role of health-related users in the generation of content, (ii) to analyze whether there is a relationship between the validity and quality of the information found in the videos and the presence of scientific evidence in them, (iii) to assess the possible appearance of sources of misinformation for patients, and (iv) to determine the types of videos that exist in the analyzed network and their relationship with the perceived validity and quality.

## 2. Methods

### 2.1. Study Design and Ethics

The research design is comprised of an observational, retrospective, cross-sectional, time-limited study analyzing activity on the social network YouTube.

This study was considered exempt from ethical review because it was performed upon a social network, and it did not involve any patient or human data beyond measuring the Internet activity among YouTube users. In addition, this study only used data from users who consented to YouTube making their data publicly available (i.e., no privacy settings were selected by them). However, accounts of individual users have been anonymized to develop good research practices on social media [43].

### 2.2. Data Collection

The data extraction system used in the present study has been through an API (Application Programming Interface) search tool, using the professional version of the software NodeXL (Social Media Research Foundation).

To achieve the objectives proposed in this study, the criteria to select the videos in YouTube were: (i) the keywords “real food”, “realfood” and “cancer” and the hashtags #realfood and #cancer were selected. (ii) videos in English. (iii) videos available on 1 December 2022. The exclusion criteria were: (i) non-English videos, (ii) advertisements, (iii) videos not related to real food and cancer in humans.

### 2.3. Data Analysis

The analysis of the data obtained was performed in several steps. The first step was collecting a total of 3817 videos. Second, both the titles and descriptions of the videos obtained were analyzed to assess whether they addressed the subject matter of the proposed study. Subsequently, the ViewRatio [27] of the resulting videos was calculated and the first 100 clips were selected [40]. Finally, a detailed analysis was carried out by viewing the videos, and it turned out that 28 of them did not deal with anything related to the subject matter required in the study (Figure 1).

This analysis was conducted by two researchers (S.S.-F. and M.d.C.L.-E.) and then corroborated by a third one (P.J.J.H.). The videos were reviewed by a group of experts including physicians, nurse, and a pharmaceutic-nutritionist, so that any differences in approach and focus were always discussed and resolved with full agreement. The following data was retrieved: upload date, number of views, number of likes.

Moreover, two indexes were calculated to compare the videos with each other; (i) the View Ratio (number of views/days from the upload to the moment of the data collection), (ii) the Viewers interaction (number of likes + comments/number of views) [27].

Likewise, an analysis was carried out to determine whether the videos provided scientific evidence or not. To meet this objective, the videos were described to include references to scientific articles or reliable technical documents that support or confirm what was explained in the video.

The videos were scored and sorted, using the modified DISCERN instrument, which allows the classification of the quality of health information related to treatments provided in videos [44,45]. It is an instrument consisting of five items rated on a Likert-type scale from 1 (poor quality) to 5 (high quality). Where, video scores > 3 points indicate “good reliability/quality”, a score of 3 points indicate “moderate”, and scores < 3 points indicate “poor reliability” and should not be used by patients [45].

The Global Quality Scale (GQS) score was used to assess the overall quality of the video. It is a five-point scale based on the quality and ease of use of online information [27,44]. The videos were categorized depending on their content and classified according to their health information quality, looking for the presence of scientific evidence in their messages, links to reliable health organizations, and identification of the authors as healthcare professionals or reliable organizations. 

Finally, an analysis of users’ account descriptions was performed. Regarding users’ uploaded videos, the description was analyzed looking for an identification as (i) Health-related users (HRU) (government/university channels/healthcare professionals) and (ii) Non-health-related users (NHRU) (communication media, news Internet channels, individual users). The videos were also classified according to: (i) the advocacy of real food as a treatment for cancer and the abandonment of medical treatment, (ii) whether the video had a focus on testimonials claiming a cancer cure linked to real food or rather, it had an informational approach; (iii) whether the video description had links to external sources that allowed verification of what was stated in the video.

### 2.4. Statistical Analysis

Descriptive and inferential statistics for analysis were performed via the Statistical Package for the Social Sciences software (SPSS) version 23.0 (IBM, Armonk, NY, USA). Descriptive statistics are presented, medians were used for quantitative variables and proportions were used for qualitative variables. Spearman’s nonparametric correlation coefficient (Spearman’s Rho) was used for correlational analysis. Mann Whitney’s U was used to compare the numerical variables. Multivariate linear regression was used to characterize relationships between video characteristics, upload source, content category, reliability (DISCERN), and educational quality (GQS). The statistical significance level was set at *p* < 0.05.

## 3. Results

### 3.1. Description of the Sample

Of the 72 videos selected after review by the researchers, the total number of views was found to be 44,682,055. Each video was viewed a total of 620,584.09 (CI95%: 177,449.33–1,068,942.71) times (Table 1). Regarding the remaining totals, it was found that 32,956 comments were obtained, with 708,351 likes and only 191 dislikes. 

The DISCERN value in the total number of videos viewed was 2.25 (±0.88) points, indicating low reliability. A similar situation was observed when analyzing the GQS, with an average value of 2.208 (±1.11). Values below 3 indicate a low quality of the information, as well as difficulty in contrasting the statements found in the videos.

When analyzing the videos in terms of scientific evidence, it is found that the average DISCERN for videos with scientific evidence is 3.61 (0.67) and video without scientific evidence was 2.09 (0.693). The GQS shows a value of 3.11 (1.17) to videos with scientific evidence and 2.22(1.02) to videos without scientific evidence, finding that the validity and quality are perceived as better in videos with scientific evidence. Regarding the geographical location of the channels where the analyzed videos were posted, it was found that 56 (77.78%) were in the USA, UK (6; 8.34%), India (4; 5.55%), Canada (4; 5.55%), and other countries (2; 2.78%).

Regarding the categorization, it was found that the videos uploaded by HRU represented only 20.8% (n = 15) of the total number of videos analyzed. Of these, 12 (16.66%) correspond to users defined as physicians, 2 (2.77%) belong to hospitals, and 1 (1.38%) to a research center. Of the 57 videos categorized as non-health related, 38 (52.8%) were found to correspond to Blog Channels, 13 (18%) were broadcast by individual user accounts (youtubers), and 6 (8.3%) corresponded to television channels.

It was also observed that the videos that offered only testimonials represented a total of 31.9% (n = 23). Of these, only 5 videos (21.74%) were from HRU.

On the other hand, videos suggesting that the use of foods defined as “real food” could cure cancer without the intervention of any other treatment accounted for 12.5% (n = 9) of the total, while the remaining videos (63; 87.5%) stressed the need to treat cancer with conventional medical treatments. All videos that presented real food as the one valid treatment came from blog channels (n = 7; 77.7%) and youtubers (n = 2; 22.22%).

Likewise, videos that provided external links to scientific/technical evidence verifying the information represented only 13.89% (n = 10) of the total number of videos (Table 2). Of these videos, 7 corresponded to HRU (70%) versus 2 that came from Blog Channels and 1 from a TV channel.

The DISCERN value for videos from HRU users was 3.05 (0.88), a value that reflects a good reliability of videos from these users. However, the GQS score is 2.8 (1.02), indicating moderate quality. 

Finally, we analyzed the different kinds of cancer treated in the videos selected. It could be observed that 19 videos (26.39%) were focused on a specific kind of cancer meanwhile 53 (73.61%) treated cancer in a generic way.

From these 19 videos, the cancers mentioned were: (i) breast cancer with 9 videos (47.37%), (ii) prostate cancer with 4 videos (21.05%), (iii) colon cancer with 2 videos (10.53%), (iv) ovarian cancer with 2 videos (10.53%), (v) pancreatic cancer 1 video (5.26%), and (vi) melanoma with 1 video (5.26%)

### 3.2. Analysis According to the Type of Videos and Users

When analyzing the results with respect to the type of video found, no significant statistical difference was found between the testimonial-type videos and those with an informational approach, for any of the categories analyzed (Table 3).

However, among those videos stating that foods classified as “real food” can be used as a legitimate treatment, it was observed that there was a statistically significant difference in favor of videos underlining that the conventional medical treatment should be continued, and never abandoned (U = 165; *p* = 0.028), with no difference found in terms of views, video duration, time since upload, likes, dislikes, comments, views, views ratio, or GQS.

The analysis of the typology of users shows that HRU presented a significant difference in likes versus those NHRU (U = 260; *p* = 0.02), as well as in the View Ratio (U = 233; *p* = 0.007). Finally, a significant difference was also observed in reliability (DISCERN), between videos of HRU versus those who were not (U = 282; *p* = 0.028). No statistically significant differences were observed in the rest of the variables analyzed between the two groups.

Between the videos with external links to scientific evidence and those without significant differences were found in favor of videos with links both in terms of reliability (DISCERN) (U = 86.5; *p* = 0.0001) and quality of the information provided (GQS) (U = 159; *p* = 0.011). For the remaining variables, no significant differences were found between the two groups. 

Likewise, we compared the reliability (DISCERN) and quality of information (GQS) of videos that treat specific cancer types. It was observed that when comparing the validity of the videos, those that dealt with cancers in a specific way offered a higher level of confidence (U = 366; *p* = 0.017). However, the quality of the videos showed no difference between the two types of videos (U = 437; *p* = 0.114).

It should be noted that, when assessing the different types of cancers and whether the videos had scientific evidence, it was found that: (i) breast cancer only had three videos out of nine that offer evidence (33.33%). (ii) In prostate cancer, three out of four videos offer evidence (75%). (iii) In both colon and ovarian cancer, there are two videos of which only one offers scientific evidence (50%). (iv) Finally, it was observed that the video on melanoma does not offer scientific evidence, while the video on pancreatic cancer does offer solid scientific evidence. 

In the remaining videos, where cancer is addressed in a general way, there is only one video that offers scientific evidence (1.89%)

### 3.3. Correlation Analysis between Popularity Indexes, DISCERN and GQS

The study of the possible correlations between the variables associated with video popularity in terms of reliability (DISCERN) and quality (GQS) (Table 4) shows that there is a positive correlation of medium intensity between video reliability and likes (r = 0.245; *p* = 0.038), comments (r = 0.266; *p* = 0.024), view ratio (r = 0.353; *p* = 0.002).

However, when assessing the quality of the video (GQS), it was observed that there was no statistically significant correlation between the GQS score and the different popularity variables analyzed (Table 4).

## 4. Discussion

This study analyzes the reliability of YouTube videos related to the real food movement and its impact on the development of cancer, either at the level of prevention -accompanying medical treatment- or even as a proposal to use only food as a treatment for cancer. YouTube is the world’s most widely used social network based on video sharing. Anyone can share information for free after registering. This has made the platform an important source of health information, especially during the pandemic, when the consumption of videos related to health problems and treatments increased exponentially [46,47]. However, it is important to note that this large amount of health information represents a non-negligible potential danger, as this information is not verified after publication [46].

To answer the first objective of the study, we assessed the role of users in generating content on YouTube channels. It was found that this information is mainly provided by accounts or users who do not identify themselves as healthcare professionals or health institutions. This trend can be observed in other studies analyzing YouTube as a source of information for pathologies such as diabetes [27], cancer [3,5,9,18], dermatological conditions such as psoriasis [25], and even vaccination against SARS-CoV-2 during the COVID-19 pandemic [47]. However, this situation is not exclusive to YouTube, as it has also been found in other social networks based on written communication, such as Twitter, also in pathologies such as cancer [20,48] and even in primary prevention strategies such as vaccination against the COVID-19 pandemic [48]. 

Previous studies analyzing the content posted on YouTube for other pathologies, such as pancreatic cancer or psoriasis, have shown that non-healthcare accounts have the highest number of likes [18,25]. An example of this is the study by Cakmak & Mantoglu in 2021 [18], where in an analysis of YouTube videos related to pancreatic cancer, it is the patients who generate the most likes compared to either health-related accounts, or those created by users defined as healthcare professionals. Similarly, Li et al., 2019 [46] described that users who watch health content on YouTube seem to like videos with low scientific quality information more than those with high quality. This situation is also observed in other studies such as the one developed by Barlas et al., 2022 [27], on myocarditis and vaccines against COVID-19 where videos with a low level of scientific evidence were also the most viewed.

This previous scenario contrasts with what was observed in our study, where the number of likes and the view ratio show a significant difference in favor of the videos provided by health-related accounts compared to the other accounts. This may be since the information provided by health-related accounts was easily recognized as relevant by the viewers or that it was sufficiently attractive, in contrast to the studies [25].

When analyzing our second objective, which was to explore the possible relationship between the validity and quality of the information found in the videos and the provision of scientific evidence, we observed that the information conveyed in the analyzed videos had a very low reliability. In addition, it is very difficult to corroborate the information provided. These results confirm the conclusions of several studies on the low quality of the information communicated through YouTube in different pathologies [49,50]. In addition, a result of great interest found in this study is that the videos that focused on treating cancers specifically were those that offered a higher level of scientific evidence and were also those that were perceived as having more validity for users. 

Most of the content shared in the videos came from users who did not identify themselves as healthcare professionals, and only a small portion was shared by these professionals. As seen in previous studies, the information they provide is more reliable, and they are also the ones who provide most external links to scientific evidence [18,44].

In this case, the percentage of videos uploaded by health-related accounts is only 20.8%, while the rest of the videos are shared by other accounts unrelated to this field. This data is particularly noteworthy when compared to previous studies analyzing the relationship between YouTube and other pathologies, in which a large part of the videos was shared by accounts related to the health world [18,25]. We can conclude that there is much more intrusiveness when it comes to sharing content on YouTube related to food and cancer. 

Regarding the third objective of this study, to analyze the possible emergence of sources of misinformation for patients, it is noteworthy that there is a significant difference in favor of videos that recommend never abandoning conventional treatment. This kind of videos are shared by accounts related to the healthcare world, which in turn are the ones that provide the most external links to scientific evidence. Since these recommendations are based on verified information [37,38], we can establish an association between the quality of the information in the videos and the presence of scientific evidence.

We also find videos that directly misinform about cancer treatment, such as those claiming that real food can cure this disease [28,29]. On the other hand, health-related users highlight the importance of nutrition and diet in the development of cancer [35,36] in parallel with antineoplastic treatments [37,38]. Some of them emphasize the importance of avoiding the consumption of ultra-processed foods which can be harmful to health [33,36,40].

These users who create sound content linking real food and cancer attempt to refute the unverified information that suggests the consumption of certain foods as the only treatment for cancer [28,29] and even referring to them as “anti-cancer” or “cancer-busting” foods [29]. 

Regarding the fourth objective, which seeks to know the existing types of videos, we do indeed find these videos recommending the intake of the so-called real food as the only treatment for cancer [28,29], disregarding any other type of treatment. Their presence confirms what other studies have previously concluded, about the use of social media to promote treatments that have already been shown to be ineffective [51,52]. However, this study yields a significant difference in favor of videos stating that medical treatment should never be abandoned.

On another note, the study has several limitations mainly related to the design, since one of the characteristics of Internet analysis in general, and YouTube in particular, is that the content is constantly changing, which means that a cross-sectional design cannot be applied. Moreover, conducting the study in one single social network, YouTube, is a limitation that must be considered, since it is possible that the topics of real food and cancer can be addressed in other audiovisual social networks, such as TikTok or Instagram. Finally, since specific keywords and hashtags were used to retrieve the information, it is possible that some videos that do not use this combination of keywords in their description and/or title may have been lost. 

An interesting line of research that can be explored following the results of this study is to investigate why in some YouTube-based studies the information provided by health-related accounts becomes relevant in both views and likes, while in other cases it is the unverified information that stands out [25,46].

On the other hand, this study presents great strengths, being one of the few to conduct such a comprehensive analysis of real food and cancer on YouTube. The chosen social network is the most popular in the world in terms of video posts. Its content was analyzed, but also the repercussions it generated, which are very relevant findings when it comes to understanding the way information travels across the Internet. 

## 5. Conclusions

Social network is one of the most relevant tools for transmitting information today. The ease of access and the usefulness of conveying the message in the form of a video or audio make YouTube one of the main sources of data transmission in the world. This also means that there is a large amount of unverified information being shared by people who do not identify themselves as healthcare professionals and do not base their statements on any scientific source. This situation can affect people’s health and become a public health problem.

In this study, a search was conducted for information related to “real food” and its association with cancer. Due to the ease of sharing content on social media, it was expected that there would be videos with unverified or incorrect information. However, as we conducted the study, we found that the conclusions go far beyond that, to the point of some users sharing information that can be harmful to patients. Videos have been found going so far as to recommend stopping antineoplastic treatment just to focus solely on a superfood approach. 

One of the characteristics of social networks is that, to date, there are no health-related content filters implemented by the platforms themselves, so any user can share a video, including those that have no scientific basis. These videos offer recommendations or spread information, that expose contradictory messages, compared with the information that patients and their families receive from physicians, nurses, and other healthcare professionals. Even when this type of videos is detected, when it is done, since it is very complicated to find them due to the huge amount of existing videos, it is a very complex process by which the video could be removed from the social network to prevent it from continuing to be displayed. 

As this study shows, it is a public health problem that is being ignored by the companies allowing this kind of content, and by the public administrations that do not act. It is important that measures are set in motion to prevent the dissemination of unverified material, or at least establish some kind of verification by the scientific community labeling which videos are based on evidence, so that when a person turns to this platform, they quickly know if what they are watching is verified content or, on the contrary, if they are voluntarily consuming content that has no quality control. 

This is the key, and the danger of this situation is that viewers do not know what they are watching. With a simple “check” on the videos, we could make the population aware of the information they are about to consume. If they still choose to consume unverified videos, they do so on their own accord. But right now, people do not know if what they are about to consume is or is not verified content. And in that case, whose responsibility, is it? Is it only the responsibility of the individual who does not know the scientific method, or is it also the responsibility of the platform that allows the content, and even of the institutions that overlook the situation?

But not everything we found in this study is negative. Videos shared by health-related accounts with verified data do have a greater impact than videos uploaded by non-health-related accounts. This is an essential element in the fight against misinformation on social networks. The upside is that in the case of real food and cancer, verified information is more relevant, despite having a lower number of shared videos than those without scientifically valid facts. 

Faced with this situation, where platforms and institutions do not act, it seems to be of utmost importance that healthcare professionals understand the need to be more present in social media from a professional point of view. In this way, they can become key figures in the creation and dissemination of reliable information from a scientific point of view, aimed at health care.

Videos that misinform will continue to exist unless action is taken, so it is critical that healthcare professionals begin to engage in the use of social media, identifying themselves as such, so that they can serve as a reference for other users. In the absence of external scientific verification, the best thing that healthcare professionals can do is to provide content of such high quality that it eclipses everything else.

## Figures and Tables

**Figure 1 ijerph-20-05046-f001:**
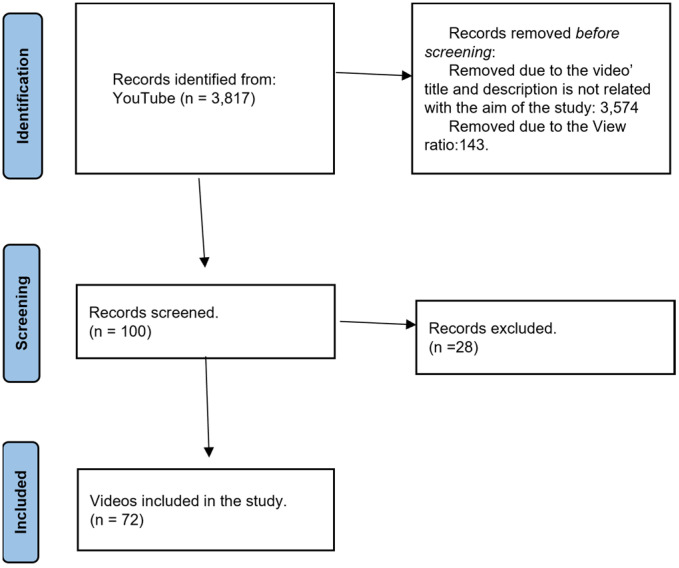
Decision flow chart, based on PRISMA flowchart.

**Table 1 ijerph-20-05046-t001:** Description of the questions that make up the reliability (DISCERN) and global quality (GQS) of YouTube videos.

	**Questions**
**Modified DISCERN**	Are the explanations given in the video clear and understandable?
	Are useful reference sources given? (Publication cited, from valid studies)
	Is the information in the video balanced and neutral?
	Are additional sources of information given from which the viewer can benefit?
	Does the video evaluate areas that are controversial or uncertain?
	**Criteria**
**GQS score**	Poor quality, poor flow, most of the information missing, not helpful to patients.
	Generally poor, some information given but of limited use to patients
	Moderate quality, some important information is adequately discussed
	Good quality, good flow, most relevant information is covered, useful for patients
	Excellent quality and flow, very useful for patients.

**Table 2 ijerph-20-05046-t002:** Summary of characteristics of YouTube videos included in the study.

			Interquartile Rank	
	N	Median	25th	75th
Duration (min)		7.55	4.02	33.47
Time from upload (days)		1090.00	707.50	1692.75
Likes		470.50	91.25	5406.00
Comments		42.50	5.00	262.00
Views		26,235.50	6792.25	232,056.50
View Ratio		30.39	5.74	379.46
VI		0.016	0.0107	0.027
DISCERN		2.00	2.00	3.00
GQS		2.00	1.00	3.00

Where: min, means minutes. VI, means viewers’ interactions.

**Table 3 ijerph-20-05046-t003:** Description of video features by category.

		Total	Video Type		Treatment		Type of Users		Scientific Evidence (in Video)	
			Divulgation	Testimonial	Yes	No	HRU	NHRU	No	Yes
Videos	N		49 (68.05%)	23 (31.95%)	63 (87.5%)	9 (12.5%)	15 (20.83%)	57 (79.17%)	62 (86.11%)	10 (13.89%)
Duration	Total	1369.1	862.81	506.32	1188.63	180.5	336.7	1032.43	1012.9	356.23
	Median (I.R.)	7.55 (4.02–33.47)	8.1 (5.4–28.4)	6.19 (3.105–36.38)	7 (4.31–33.65)	8.28 (3.5–25.55)	7 (5.795–35.95)	8.1 (3.4–18.3)	7 (3.675–18.2675)	31.2 (6.242–58.55)
	U; *p*-value		535; 0.731		276; 0.905		367; 0.401		213.5; 0.118	
Time from upload (days)	Total	82,689	54,209	28,480	72,984	9705	18,933	63,756	71,407	11,642
	Median (I.R.)	1090 (707.5–1692.75)	1080 (700–1500)	1200 (820–1750.5)	1085 (720–1600)	1100 (395–1701)	1200 (955–1905)	1080 (700–520)	1082 (702.5–1775.25)	1147.5 (915–1477.5)
	U; *p*-value		486; 0.349		264; 0.746		356; 0.325		294.5; 0.807	
Views	Total	44,682,055	28,897,853	15,784,202	42,182,482	2,499,573	24,238,349	20,443,706	410,107,746	3,671,309
	Median (I.R.)	26,236	66,105 (5215–496,672)	15,201 (7641–96,163)	33,249 (6860.5–235,426)	8120 (1686–32,596)	228,687 (47,491–993,376.5)	16,093 (4467–102,236)	21,265.5 (5564.5–238,795.5)	76,612.5 (14,605.25–128,985)
	U; *p*-value		482; 0.328		214; 0.24		226; 0.005 ***		269.5; 0.515	
Comments	Total	32,956	20,422	12,534	26,196	6760	15,120	17,836	28,860	4096
	Median (I.R.)	42.5	62 (5–261)	36 (5.5–233.5)	37 (5.5–263)	54 (3–160)	295 (14–850.5)	37 (5–160)	37.5 (5–264)	82.5 (11.5–151)
	U; *p*-value		563; 0.995		279; 0.939		308; 0.097		299.5; 0.87	
Likes	Total	708,351	543,299	165,052	652,538	55,813	311,991	396,360	617,816	90,535
	Median (I.R.)	471	862 (80–11,634)	372 (136–2370)	485 (102.5–5464)	351 (18–929)	5348 (330–20,678)	372 (63–3811)	410 (83.75–4716)	1497.5 (473–10,120.5)
	U; *p*-value		520; 0.604		243; 0.49		260; 0.02 *		249.5; 0.329	
View Ratio	Total	51,346.3	37,670.271	13,676.005	41,600.179	9746.097	22,383.396	28,962.88	48,163.234	3183.042
	Median (I.R.)	30.387 (5.746–379.46)	49.871 (5.762–469.978)	15.978 (6.793–81.933)	44.07 (6.53–386.36)	11.123 (1.36–40.74)	371.725 (43.033–1436.07)	16.046 (4.211–127.88)	17.305 (5.612–393.261)	54.79 (21.364–68.939)
	U; *p*-value		482; 0.33		213; 0.233		233; 0.007 **		276; 0.585	
VI	Total	n.a	n.a	n.a	n.a	n.a	n.a	n.a	n.a	n.a
	Median (I.R.)	0.0162 (0.0107–0.0272)	0.0156 (0.0106–0.024)	0.0178 (0.012–0.033)	0.0151 (0.0105–0.025)	0.0221 (0.0149–0.0431)	0.0176 (0.012–0.0273)	0.0146 (0.00915–0.0253)	0.015 (0.0106–0.024)	0.0231 (0.0154–0.0364)
	U; *p*-value		457; 0.464		262; 0.202		339; 0.222		232; 0.207	
DISCERN	Total	n.a	n.a	n.a	n.a	n.a	n.a	n.a	n.a	n.a
	Median (I.R.)	2.00 (2–3)	2.00 (2–3)	2.00 (1.5–2)	2 (2–3)	2 (1–2)	2 (2–3)	2 (2–3)	2 (2–2)	3 (3–3.75)
	U; *p*-value		426; 0.07		165; 0.028 *		282; 0.028 *		86.5; 0.0001 ***	
GQS	Total	n.a	n.a	n.a	n.a	n.a	n.a	n.a	n.a	n.a
	Median (I.R.)	2 (1–3)	2.00 (1–3)	2.00 (1–2.5)	2 (1–3)	1 (1–3)	2 (1–3)	2 (1–3)	2 (1–3)	3 (2.25–4)
	U; *p*-value		477; 0.276		191; 0.101		407; 0.767		159; 0.011 **	

Where: VI, means viewers’ interactions; I.R.; means interquartile rank. Treatment refers to whether the videos suggested that the exclusive use of real foods could cure cancer. n.a.; means “not applicable”. * *p* < 0.05; ** *p* < 0.01, *** *p* < 0.001.

**Table 4 ijerph-20-05046-t004:** Correlation between DISCERN and GQS with popularity indexes.

	DISCERN	GQS
	r; *p*-Value	r; *p*-Value
Likes	0.245; 0.038	0.12; 0.314
Comments	0.266; 0.024	0.142; 0.235
View Ratio	0.353; 0.002	0.204; 0.085
VI	0.079; 0.512	0.094; 0.433

## Data Availability

The data that support the findings of this study are available from the corresponding author upon reasonable request.

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
