# Peer review of "Realfood and Cancer: Analysis of the Reliability and Quality of YouTube Content"

_ijerph, 2023, doi:10.3390/ijerph20065046_

Round 1

Reviewer 1 Report

A literature review should be added to contextualise the study, or worked into the introduction – it doesn’t need to be extensive but it should mention the established research on social media and credibility / accuracy of sources, and the use of social media in the context of public health or health; after which the significance of this study would be emphasised.

Minor editing required eg. under methods 2.1 change the following to active voice – instead of  “An observational, retrospective, cross-sectional, time-limited study analyzing activity on the social network YouTube was proposed” I’d say something like “The research design is comprised of an observational, retrospective, cross-sectional, time-limited study analyzing activity on the social network YouTube”. Under Discussion, this sentence could be more precise-“This study offers an analysis of the reliability of YouTube videos…” should be changed to “This study analyses the reliability of YouTube videos…” Under Conclusions, get rid of informal phrasing “…heeding this recommendation can be terrible for the patient” and “But oddly enough…” inappropriate for an academic publication, and small general expression issues throughout eg. “the companies allowing this kind of contents”.

Author Response

Reply to reviewer:

Thank you very much for your contributions. It is certainly essential to include in a more specific way in the introduction the importance for public health of the issues addressed in this work. So, we have modified the introduction to include this type of information for future readers.

Likewise, we have modified everything related to the indicated editing changes. We have rewrite theses senteces to improve the understanding of the text and avoiding possible colloquial expressions.

Reviewer 2 Report

This paper describes an interesting analysis of videos about real food and cancer on Youtube platform. The report is well-structured and easy to follow. However, a few aspects need improvement. Please see more detailed suggestions below.

1.Who exactly evaluated the videos? With what specialty? 

2. The authors analyzed the reliability and quality on real food and cancer. In the final 72 videos, What are the types of cancer? Is any relationship between cancer type and reliability and quality? 

3. What are the limitations of this study?

4.Minor comments

----------------------------

Abstract

Sentence: “The videos uploaded by HRU represented only 20.8%.”  What does the abbreviation of HRU stand for?

Author Response

This paper describes an interesting analysis of videos about real food and cancer on Youtube platform. The report is well-structured and easy to follow. However, a few aspects need improvement. Please see more detailed suggestions below.

1.Who exactly evaluated the videos? With what specialty? 

Reply to reviewer: 

Thank you very much for your comment, we have included the information about the reviewers in the text so that it can result in a better understanding of the method followed for future readers.

  1. The authors analyzed the reliability and quality on real food and cancer. In the final 72 videos, What are the types of cancer? Is any relationship between cancer type and reliability and quality?

Reply to reviewer:

Thanks for your comment, effectively the requested data enrich the article.

The requested data has been incorporated, defined the videos that addressed specific cancer or mentioned cancer in a general way.

Likewise, we have proceeded to analyze the possible relationship in terms of the reliability and quality of the contents between the two types of videos.

These data have been included in the results, so has been included in the discussion too,  due to their importance must be mentioned there. 

  1. What are the limitations of this study?

Reply to reviewer:

Thanks for your comments. We have included the limitations as a part of the discussion, you could find them in page 11 of the manuscript

Reviewer 3 Report

Authors enlightened the role of "nutrition" in the cancer field through the evaluation of uploaded YouTube videos on the topic of interest. Despite nutrition in cancer patients remaining a hot topic, the paper presents several lacks to be addressed. Firstly, the paper does not report a PRISMA statement. It could be useful to know which are the criteria for selecting 72 videos such as other authors highlighted (PMID:36679937). How many videos are searched for each keyword? Why have they used just these keywords ("real food", “realfood” and "cancer"). Moreover, the methodology used is biased. How could they use the "Video power index" or "video like ratio" if YouTube had disabled the "dislike" until March 2022??? The authors should clarify this aspect. Additionally, how was the misinformation evaluated? How many questions have the authors postulated?.. Did the Authors subgroup analysis??? It could be interesting to know if the misinformation or low rate of DISCERN were present in a video with scientific evidence rather than one without a scientific basis. Authors should hardly improve the paper for making it suitable for publication. A major revision is required.

Author Response

Authors enlightened the role of "nutrition" in the cancer field through the evaluation of uploaded YouTube videos on the topic of interest. Despite nutrition in cancer patients remaining a hot topic, the paper presents several lacks to be addressed. 

Firstly, the paper does not report a PRISMA statement. It could be useful to know which are the criteria for selecting 72 videos such as other authors highlighted (PMID:36679937). How many videos are searched for each keyword? Why have they used just these keywords ("real food", “realfood” and "cancer"). Moreover, the methodology used is biased. 

Reply to reviewer:

We have clarified the methdologý to explain clearly how to select the videos. Finally, we are interested in analyzing the role of the real food movement and its effect on cancer treatment. 

This is due to the impact that we have observed in Spain about the change in behaviors regarding treatments and medical recommendations 2from people that selfname as gurús of the use of real food.

We don’t consider that our analysis could be defined as biased, due to our main goal is to analyze the effects of real food and how this is expressed and disseminated through Youtube, and other social networks, and ee define clearly this goal in the manuscript

How could they use the "Video power index" or "video like ratio" if YouTube had disabled the "dislike" until March 2022??? The authors should clarify this aspect. 

Reply to  reviewer:

Thank you very much for your indications and the warning for the use of 2 inadequate indicators.

As the reviewer has indicated, this approach has been inadequate, and an error to include it. To solve this situation we have eliminated those indexes that are perverted by not being able to count on the measure of dislikes.

Likewise, all reference to them has been eliminated in methodology, results, and discussion.

Additionally, how was the misinformation evaluated? How many questions have the authors postulated?.. Did the Authors subgroup analysis??? It could be interesting to know if the misinformation or low rate of DISCERN were present in a video with scientific evidence rather than one without a scientific basis. Authors should hardly improve the paper for making it suitable for publication. A major revision is required.

Reply to reviewers

Thank you very much for your comments, as you have rightly expressed we had not clearly indicated what was valued as scientific evidence and the access to it.

We have included a description in the methodology that defines the approach given to what is considered scientific evidence included in a video. 

An analysis has been included in the results, which assessed the level of reliability and quality of the videos according to a new subgroup included in the study, the cancers addressed in the videos analyzed.

likewise, new information has been included in the results in response to the question asked, in which the value of the average is requested to see the value of this questionnaire. Finding that the validity and quality are perceived as better in videos with scientific evidence than without scientific evidence..

Round 2

Reviewer 3 Report

Authors should add more values to the topic addressed. The revised manuscript is still scant. I advice to read 10.3390/ijerph20064721 to improve the topic presentation and introduction section.

Author Response

Reply to reviewer

Thank you very much for your comments, in the previous round of review we did not realize that we had not adequately described some points.
For example, to facilitate understanding of the review process, we have included a flowchart based on the PRISMA flowchart.
The introduction has also been modified to try to better explain the purpose of this study.